# Auditory cortical activity drives feedback-dependent vocal control in marmosets

Steven J. Eliades[1] & Joji Tsunada[1]

Vocal communication is a sensory-motor process requiring auditory self-monitoring to correct errors and to ensure accurate vocal production. When presented with altered speech feedback, humans rapidly change their speech to compensate. Although previous evidence has demonstrated suppression of auditory cortex during both speech and animal vocalization, the specific role of auditory cortex in such feedback-dependent control is unknown. Here we show the relationship between neural activity in the auditory cortex and feedback-dependent vocal control in marmoset monkeys. We demonstrate that marmosets, like humans, exhibit feedback control of vocal acoustics. We further show that feedback-sensitive activity of auditory cortex neurons predict such compensatory vocal changes. Finally, we demonstrate that electrical microstimulation of auditory cortex rapidly evokes similar changes in vocal production. These results are evidence for a causal role of auditory cortex in vocal self-monitoring and feedback-dependent control, and have implications for understanding human speech motor control.

[1] Auditory and Communication Systems Laboratory, Department of Otorhinolaryngology: Head and Neck Surgery, University of Pennsylvania Perelman School of Medicine, Philadelphia, PA 19104, USA. Correspondence and requests for materials should be addressed to S.J.E. (email: seliades@med.upenn.edu)

Vocal communication plays an important role in the lives of both humans and many animal species. Accurate communication, however, requires continuous self-monitoring of speech to detect and compensate for errors of production and perceived changes in vocal output[1]. This sensory-motor mechanism uses auditory feedback of produced vocal sounds to enable control and rapid correction of a variety of speech parameters, including amplitude, pitch, and formant frequencies[2–4]. The neural mechanisms underlying this vocal feedback monitoring and their contribution to feedback-dependent vocal control are poorly understood.

The auditory cortex, a key structure in auditory perception, has been suggested as a possible site for the sensory-motor comparison between efferent motor predictions of expected vocal feedback and the feedback actually observed[5]. Information from such a comparison can then be relayed to motor areas to drive changes in vocal production. Supporting such a function, recent work in both humans[6–12] and non-human primates[13–16] has demonstrated a vocalization-induced suppression of auditory cortical activity during vocal production, evident down to the single neuron level. Such suppression contrasts with more typically excitatory responses in the same auditory neurons during the playback of recorded vocalizations and other sounds[17]. However, despite this vocal suppression, recordings in primates have demonstrated that auditory cortex maintains its sensitivity to changes in vocal feedback and may, in fact, be more sensitive to such changes than during passive listening[15,16]. A few studies in humans have also found that feedback-related cortical activity correlates with subsequent vocal control[11,18]. However, a specific, causal role for auditory cortex in feedback-dependent vocal control has not been established.

In this study, we examined whether neurons in the auditory cortex of marmoset monkeys (*Callithrix jacchus*), a highly vocal primate species, are involved in feedback vocal control behaviors. Combining chronic neural recordings from auditory cortex during vocalization with frequency-shifted vocal feedback, we demonstrate that neurons in the auditory cortex predict compensatory vocal behaviors, and using electrical microstimulation we provide direct evidence that auditory cortex activity can evoke changes in vocal production.

## Results

**Marmosets exhibit feedback-dependent vocal control**. We recorded neural activity from the auditory cortex of two marmosets using implanted multi-electrode arrays while they produced voluntary, self-initiated vocalizations. We first evaluated whether or not marmosets exhibit feedback-dependent control of their vocalizations using triggered real-time frequency shifts of ±2 semitones (ST) to alter vocal feedback (Fig. 1a, b) and compared mean frequency contours within individual vocalizations before and during frequency-shifted feedback (Fig. 1c). Similar to the behavior expected from human studies, we found compensatory changes in vocal production, with significantly increased vocal frequencies during negative feedback shifts, and decreased frequencies during positive shifts (Fig. 1d). These compensatory changes were present in both marmosets tested, with magnitudes of mean compensatory changes (30.2–84.3 Hz, or 8.3–19.5 cents in log scale [1ST = 100 cents]) on par with those observed in the human pitch-shift reflex[2,19]. We did not see any changes in vocal amplitude or duration due to shifted feedback. We further examined the time course of vocal control by averaging vocal frequency contours, and found significant compensation beginning 215 ms following shifted feedback onset (Fig. 1e). Onset of compensation was earlier for +2 ST (135 ms) than for −2 ST (244 ms), though the origin of these differences is unclear. The

latency (~200 ms) and peak magnitude (−107 and 151 Hz, or −19 and +26 cents) are similar to those reported in human subjects. Such compensatory vocal behavior, though well studied in humans, has not been previously found in non-human primates, or any other mammalian species save echolocation sounds of bats[20], and is evidence that marmosets have feedback control of their vocalization frequencies, a behavioral effect that is highly consistent with human observations.

**Auditory cortex neurons predict vocal compensation behaviors**. In order to determine the relationship between cortical activity and vocal compensation, we recorded neural activities in auditory cortex during vocal production. Consistent with previous results[15,16], we found that suppressed auditory cortical neurons, neurons that are normally inhibited during vocalization, exhibited increased firing rates (i.e., reduced suppression) during frequency-shifted feedback, but not during amplification alone (Fig. 2a). Unlike previous studies, which used continuous feedback shifts, the reduced suppression we observed did not occur until after the onset of the triggered feedback shift (~100 ms). This feedback sensitivity was present at both the single-unit (Fig. 2a, b) and population (Fig. 2c) levels, and was similar at the population average level for both +2 and −2 ST shifted feedback.

To further identify a specific role for auditory cortex in vocal compensation, we compared neural activity during shifted feedback to changes in vocal production. One sample multi-unit recording exhibited increased firing during shifted feedback (Fig. 3a) that was greater for trials in which the animal subsequently compensated (frequency change $z$-score > 0) than trials when it failed to compensate ($z < 0$; Fig. 3b). This difference was most evident in the early feedback period (first 200 ms), preceding the average onset time of vocal compensation (>200 ms). Further individual call-by-call comparisons for this unit showed a strong correlation between firing rate and vocal compensation (Fig. 3c). Such correlation between vocal frequency change and neural activity ("vocal correlation") did not develop until the early feedback period, was gone by the onset of compensation, and was not present for non-shifted feedback trials (Fig. 3d).

Population average comparisons of neural activity and vocal compensation for suppressed units showed similar trends in firing rate and vocal correlations, including timing (Fig. 4a). Although individual units exhibited significant correlations between activity and vocal compensation, there was considerably heterogeneity amongst units, with strong correlations in both directions (Fig. 4b). Overall 8.8% of units tested with −2ST, and 8.8% tested with +2ST, exhibited significant ($p < 0.05$) correlation coefficients (when combining over call types, Table 1). Comparisons of coefficients across different call types showed little correlation ($r = 0.006$), possibly reflecting differing vocal acoustics, and there was no systematic topography to the correlations. However, the presence of units with opposite behavioral correlations, when combined, may account for reduced average population compensation differences in +2ST feedback testing (i.e., Fig. 4a). To better examine these diverse neural activities, we performed further population analysis, dividing units based upon vocal correlation (>0 vs. <0), and found that population-level compensation differences were present for both +2 and −2 ST feedback (Fig. 4c, d). These effects were stronger for units whose correlation was in the same direction as the expected vocal compensation (i.e., $r > 0$ for feedback −2ST, $r < 0$ for +2ST). Population responses showed similar timing as expected from individual units, with differences developing only during altered feedback and a normalization once vocal compensation began. Similar population responses were seen for the vocal correlation

coefficients between firing rate and subsequent vocal compensation, including magnitude and timing (Fig. 5). Collectively these results demonstrate that neural activity during altered feedback correlates with vocal compensation. Such correlation can be considered predictive as it occurs primarily in the early feedback period, prior to the onset of vocal compensation, suggesting the possible role of the auditory cortex in encoding feedback information that allows subsequent use in vocal control. Additionally, the presence of neurons with opposite predictions of vocal behavior may suggest a two-population opponent model

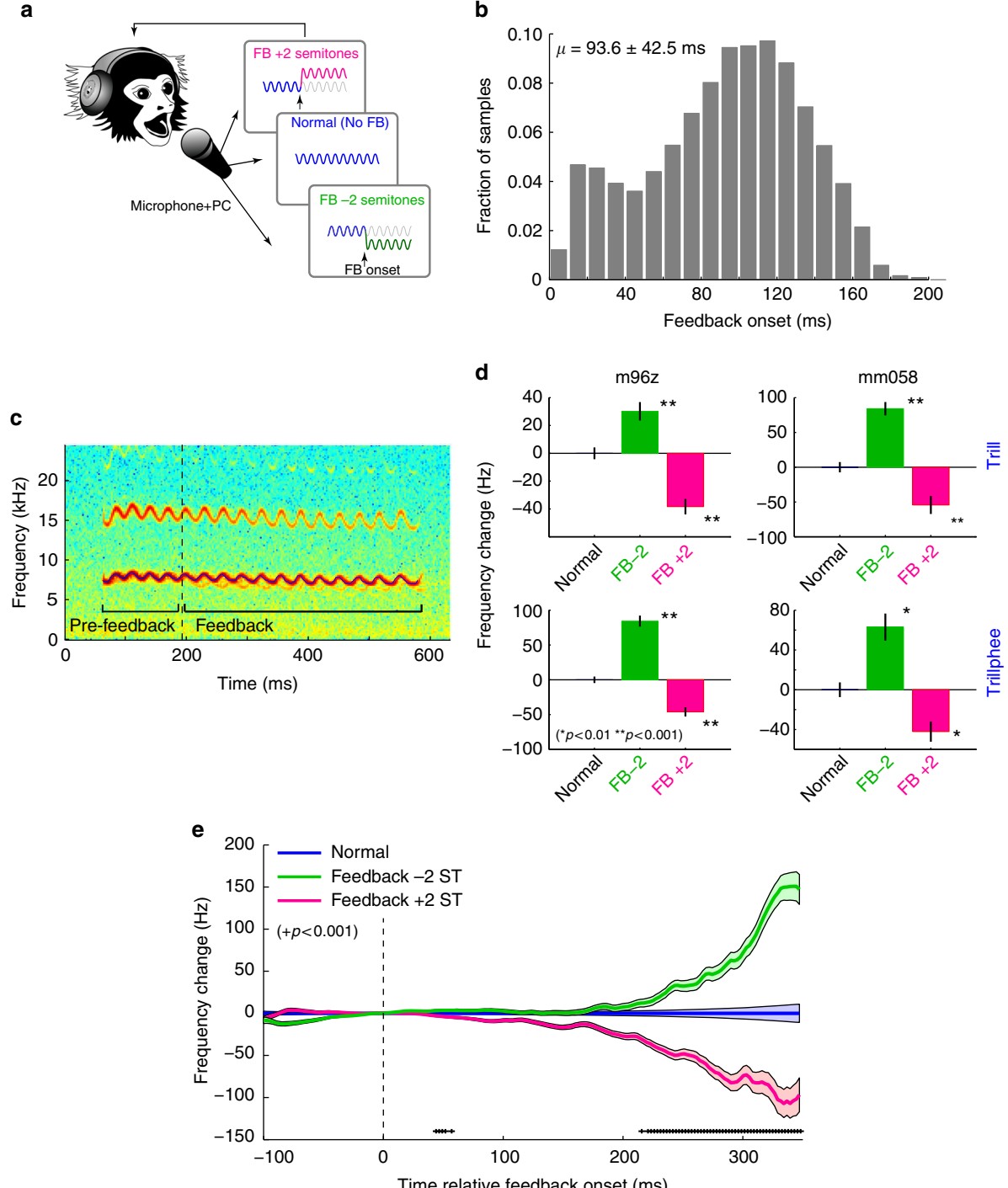

**Fig. 1** Feedback-dependent control of vocal frequency in marmosets. **a** Schematic illustration of shifted feedback experiment including +2 and −2 semitone feedback (FB) frequency shifts. **b** Distribution of feedback trigger times. **c** Sample trill vocalization illustrating −2ST (semitone) feedback including pre-feedback and feedback time ranges. The shifted feedback is weakly visible in the spectrogram. Frequency contour is highlighted (blue). **d** Comparison of mean vocal frequency change between pre-feedback and feedback time periods for normal, −2ST, and +2ST conditions demonstrating compensatory vocal changes in the opposite direction of the feedback shift. Results are shown separately for trill (top) and trillphee (bottom) vocalizations, and for the two animals. Error bars: standard error (SEM); *p < 0.01, **p < 0.001 for each feedback vs. normal (Kruskal–Wallis test). **e** Time course of compensatory vocal changes in trills during shifted feedback demonstrating expected vocal compensation with a 215 ms latency from feedback onset (Error bars: SEM, +p < 0.001, Kruskal–Wallis). Sample sizes and statistical details may be found in the Supplementary Note 1

for vocal control, i.e., groups of neurons whose activity drives vocal frequency control through activation of one group of neurons, and deactivation of the opponent neurons. Such a push-pull system would allow for bidirectional behavioral control, as we have observed for marmoset vocal production. Such opponent populations may have been less obvious in previous human studies due to limitations in the specificity of human recording methods. The origin of these neural preferences for the direction of vocal compensation, however, remains unclear.

Because vocalization-induced suppression of auditory cortex has been specifically implicated in self-monitoring and possibly compensatory behaviors, we further quantified the relationship between the degree of suppression and vocal compensation for single-unit activities. Consistent with previous results[15], a strong correlation was observed between feedback sensitivity and vocal suppression, with greater sensitivity seen in suppressed units than excited ones (Fig. 6a, b). However, we did not identify any consistent patterns between vocal suppression and predictive vocal correlations (Fig. 6c, d). Comparisons of unit frequency tuning, feedback sensitivity, and vocal correlations also did not show any consistent patterns (Fig. 6e, f). There was a small asymmetry in responses between hemispheres, with greater feedback sensitivity in the right over the left hemisphere (Fig. 6g), but no hemispheric differences were noted for vocal correlations (Fig. 6h). Analysis of medial-lateral location, likely correlating with primary vs. non-primary auditory cortex, did not show consistent patterns ($p > 0.05$ for both feedback and vocal correlation, Kruskal–Wallis), nor were there differences between the two animals. These findings confirm that auditory tuning and anatomic location alone cannot account for feedback responses or compensation and although there is a clear correlation between vocal suppression and feedback sensitivity, a specific role in compensatory vocal control remains uncertain.

### Auditory cortical microstimulation evokes rapid vocal change.
In order better determine the connection between auditory cortex and vocal control, we next introduced electrical microstimulation during vocalization to modulate the activity of the auditory cortex. We used the same electrodes from the neural recordings,

allowing us to directly compare stimulation results and neural activities. As with shifted auditory feedback experiments, stimulation was triggered to begin shortly after the onset of vocal production. Examination of raw vocal spectrograms revealed abrupt changes in vocal frequency on many trials (Fig. 7a). One common pattern was an abrupt increase and/or a sustained drop in vocal frequency. Such changes were not observed during vocalizations in the absence of electrical stimulation. The median latency of these vocal changes from stimulation onset was very rapid, 38.5 ms (Fig. 7b). Comparison of stimulation changes between electrodes revealed increases in more lateral electrodes, likely non-primary auditory cortex, as well as greater stimulation effects in the right hemisphere of both animals (Fig. 7c, d). Such a right hemispheric asymmetry was also present for comparisons of feedback sensitivity (Fig. 6g) and may be consistent with recent human results[21], as well as hemispheric differences in processing of vocal pitch and prosody[22]. However, such hemispheric differences may have also resulted from variability in the placement of electrode arrays between hemispheres.

To further examine these microstimulation effects, we performed quantitative analysis of stimulation-evoked frequency changes for each electrode, confirming the differences between stimulation and non-stimulation trials (Fig. 7e). Examination of the entire electrode grid revealed a wide variance in stimulation effects, both in terms of strength, timing, and direction (Fig. 7f). Similar patterns were seen in electrode grids of both animals (Fig. 8). The variability of stimulation effects between electrodes, and absence in some, indicate a site-specific effect rather than a non-specific result of stimulation. Interestingly, large decreases in vocal frequency were the most commonly observed stimulation effect, while increases were weaker and later (>150 ms from stimulation onset).

In order to better understand the origin of electrical stimulation-evoked vocal changes, and the relationship to vocal suppression, we compared stimulation effects to feedback neural responses recorded from the same sites. We first quantified the magnitude of the stimulation effect by the median change in vocal frequency, again showing bidirectional changes with a negative bias, and stronger effects in the right hemisphere (Fig. 9a). Multivariate linear regression for stimulation effects as a function

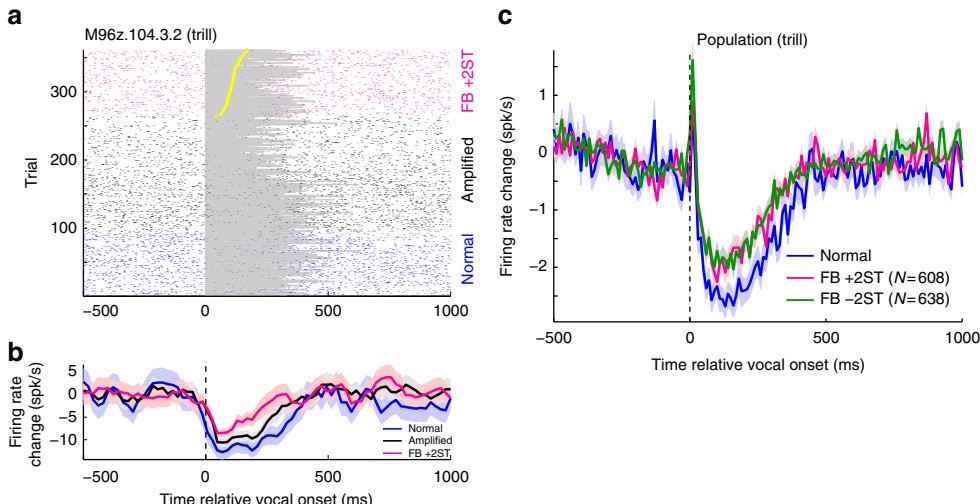

**Fig. 2** Auditory cortical activity during shifted feedback. **a** Sample single-unit response raster and peri-stimulus time histogram (PSTH, **b**) from an auditory cortex neuron suppressed by trill vocal production (vocal Response Modulation Index: RMI = −0.53) but exhibiting decreased suppression during a +2ST feedback shift (normal −10.2 spk/s, amplified −9.5, FB −6.5; $p < 0.001$ Kruskal–Wallis; Error bars: SEM. Vocal intervals (shaded) and feedback onset times are indicated (yellow marker). **c** Population average PSTHs showing reduction in vocal suppression during shifted feedback compared to normal vocalizations for suppressed units (RMI < −0.2). Indicated N is number of units. Sample sizes and statistical details may be found in the Supplementary Note 1

of animal, hemisphere, electrode column (anterior-posterior) and electrode row (medial-lateral) did not find any significant effects of animal ($p = 0.37$), column ($p = 0.48$), or row ($p = 0.45$), but still showed the effects of hemisphere ($p = 0.004$).

Comparison of stimulation and vocal neural responses showed a significant correlation across recording sites (Fig. 9b), suggesting that vocalization-induced suppression may indeed play a role in vocal control, although a similar relationship had not been noted in the neural recordings. Additional correlations were found between stimulation and both feedback sensitivity (Fig. 9c) and vocal correlation coefficients (Fig. 9d). These results show that stimulation of those neurons whose feedback sensitivity (increased firing) predicted compensatory decreases in vocal frequency ($r < 0$) also resulted in decreased vocal frequency, and that the opposite is also true. Such similarity in similarity between stimulation-induced changes and vocal changes predicted by neurons' physiologic responses suggests a common mechanism in which the stimulation activates said neurons, thereby evoking a similar behavioral outcome.

However, because such measures of neural activity are themselves potentially correlated, we also constructed a multivariate linear regression model including neural responses as well as hemisphere. The model showed a strong regression fit to the data ($R^2 = 0.303$, $F = 7.73$, $p < 0.001$), with partial correlation coefficients for vocal suppression, feedback response, vocal correlation, and hemisphere of 0.26, −0.25, 0.28, and −0.06 ($p$-values 0.025, 0.033, 0.017, and 0.61), indicating that each is independently related to vocal changes, save for hemisphere. How each contributes to vocal control, is still uncertain. The absence of an independent hemispheric effect, suggests that hemispheric asymmetry during microstimulation is likely a result of variations in neural responses to vocal feedback, rather than another unaccounted factor.

Finally we compared stimulation effects between electrodes according to their mean frequency tuning (Fig. 9e). The largest decreases in frequency were noted for neurons around 4 and 8 kHz, with increases noted at 5–6 kHz. Similar patterns were noted in both hemispheres examined independently, though reduced in the left (Fig. 9e, inset). Because the mean frequency of vocalization was $6.95 \pm 0.80$ kHz (mean ± std), these results

suggest that stimulation of frequency tuned sites just above vocal frequencies may have induced a miss-coding of a vocalization as too high in frequency, thereby evoking a compensatory decrease in produced frequency, and that stimulation of frequencies just below the call evoking the opposite effect.

## Discussion

In this study we examined the role of the auditory cortex in feedback-dependent vocal control. Combining frequency-shifted auditory feedback, chronic neural recordings, and electrical microstimulation, we have shown the direct, causal evidence of the auditory cortex's role in vocal control, and the first evidence in any species that direct manipulation of a sensory brain area can influence ongoing vocal production in real-time. We found that (1) marmosets exhibit feedback-dependent control of their vocal frequencies similar to humans in timing and magnitude, (2) neurons in the auditory cortex are sensitivity to altered feedback and predict compensatory vocal control, and (3) stimulation of auditory cortex evokes rapid changes in vocal production that are similar to neural predictions from the same locations.

Although previous work in songbirds has found that stimulation of sensory nuclei can also induce a gradual change in song structure, this occurs over times scales of days[23], in contrast to the 40 ms presented here. This difference may suggest a fundamentally different mechanism for the use of vocal feedback, one in which feedback is used on a moment-to-moment basis, rather than to maintain a stable song over the long term. Indeed the presence of short-latency behavioral reflexes for frequency control in both humans and marmosets suggest just such a role. The short latency of microstimulation effects also suggests the presence of a short neural reflex arc between auditory cortex and the larynx, possibly involving pre-motor and motor cortex based upon models of vocal motor control in both humans[5] and primates[24]. The precise neural pathway that might constitute this audio-motor projection is uncertain. In humans, the arcuate fasciculus connects the superior temporal gyrus with pre-motor areas involved in speech[25]. Similar pathways appear to be present in non-human primates, connecting the non-primary auditory cortex with pre-frontal cortex in both marmosets[26] and

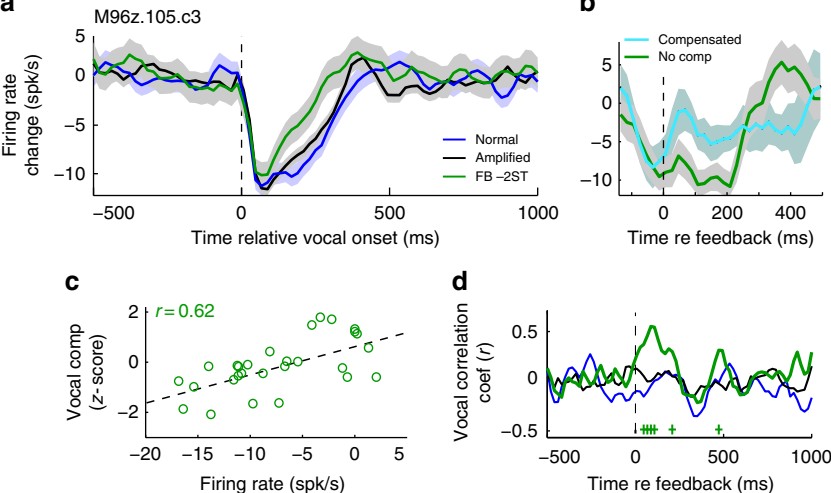

**Fig. 3** Neural prediction of vocal compensation. **a** A sample multi-unit (MUA) PSTH of a single recording site is shown in response to a −2ST shift. **b** Splitting feedback responses into compensated and no compensation trials (vocal compensation z-score > 0 vs. <0) showed an early increase in activity during compensated trials that preceded vocal compensation (which occurs >200 ms). **c** Correlation between feedback responses (0–200 ms) and future vocal compensation (>200 ms, $r = 0.62$, $p < 0.001$). Linear regression fit is shown (dashed). **d** Sliding-window analysis showed that this correlation did not develop until after feedback onset and lasted until the start of vocal compensation ($+p < 0.05$). Sample sizes and statistical details may be found in the Supplementary Note 1

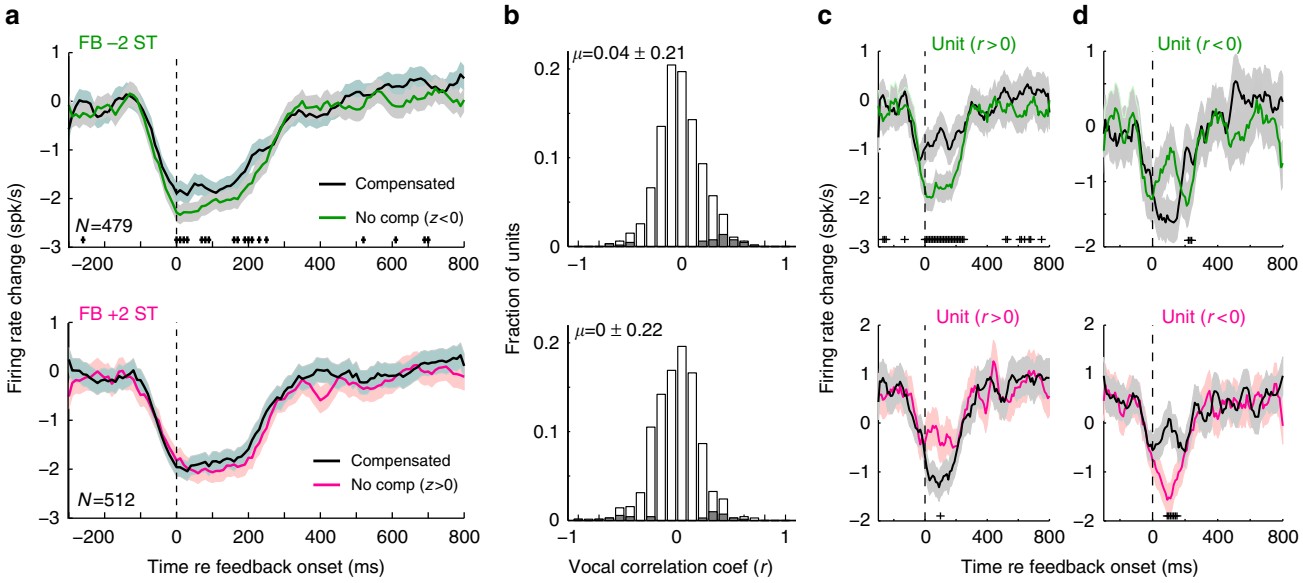

**Fig. 4** Populations responses preceding vocal compensation. **a** Population average PSTHs showing differences between compensated and non-compensation trials for FB −2ST (top) and +2ST (bottom) in suppressed units (RMI ≤ −0.2). Number of units are indicated (Error bars: SEM; +$p$ < 0.05, Kruskall–Wallis). **b** Population distribution of individual unit vocal correlation coefficients showing variability between units (Shaded: $p$ < 0.05). Mean ± std (standard deviation) indicated. **c**, **d**, Population PSTHs after splitting units between those with vocal correlation coefficients $r$ > 0 (**c**) and $r$ < 0 (**d**). PSTHs show differential activity starting after the onset of altered feedback. Differences were noted to be stronger for units whose correlation coefficients predicted the expected behavioral compensation (bolded) during altered feedback: $r$ > 0 for FB −2ST (**c**, top) and $r$ < 0 for +2ST (**d**, bottom), as evidenced by the presence of significantly different time bins ($p$ < 0.05; −2ST: 23 significant time bins in the 250 ms for $r$ > 0 vs. 3 in $r$ < 0; +2ST: 7 bins for $r$ < 0 vs. 1 for $r$ > 0). Sample sizes and statistical details may be found in the Supplementary Note 1

### Table 1 Number of units with significant vocal correlations

| Animal | Hemisphere | Feedback -2 ST | | Feedback + 2 ST | |
|---|---|---|---|---|---|
| | | **Significant** | **Total** | **Significant** | **Total** |
| m96z | Left | 2 | 56 | 13 | 75 |
| | Right | 29 | 267 | 25 | 284 |
| mm058 | Left | 10 | 120 | 4 | 111 |
| | Right | 0 | 21 | 1 | 20 |

Number of units with significant vocal correlations is shown, separated by feedback condition, animal, and hemisphere

macaques[27]. The functional role of these anatomic pathways during vocal production and control remains an open question.

Finally, the correlation between microstimulation and neural recordings is suggestive that similar neural mechanisms may have been at work in both. We have found that previously described vocalization-induced suppression[14] and self-monitoring feedback sensitivity[15] in auditory cortex precede and are predictive of subsequent vocal changes. Such observations are important supportive evidence for the possible mechanistic role of the auditory cortex in vocal self-monitoring and control. A similar correlation between activity in the auditory cortex and compensatory vocal control has also been observed in humans[11,18]. These similarities of behavioral and neural correlates of vocal control between species suggest that the mechanisms of vocal production and feedback vocal control may be conserved along the evolutionary tree. However, despite the apparent correlation between auditory cortical activity and vocal control, the specific role of vocalization-induced suppression remains unclear. Suppressed neurons exhibited greater sensitivity to feedback, as in previous marmoset studies[15], but both human studies[11] and the present data have not found a systematic relationship between suppression and predictions of vocal control. This discrepancy may

reflect the presence of two opposing neural populations, one which drives vocal frequencies upward, and one that drives downward. Such bi-directional behavior is further supported by the microstimulation results. The presence of frequency-dependence to the microstimulation provides one possible hypothesis for the origin of these opponent responses, with behavioral effects resulting from CF tuning. However, this frequency tuning dependence was not obvious in the neural responses, and may require more systematic testing of individual neurons with shifted feedback in both directions to more fully evaluate such a proposed mechanism.

It should be noted, however, that current findings are evidence that auditory cortical activity is sufficient to drive vocal control, but not that auditory cortex is necessary. Vocal/motor-induced suppression, which underlies feedback sensitivity[15], has primarily been observed at the cortical level, suggesting a cortical-cortical or thalamocortical mechanism[28]. However, multiple levels of feedback monitoring may be possible, including sub-cortical auditory areas. The roles of vocal suppression, frequency tuning, and hemisphere remain an open question, and future work will need to more specifically examine neurons in key vocal frequency ranges to better understand the mechanisms of vocal suppression and feedback sensitivity, and their specific roles in feedback-dependent vocal control. Understanding these mechanisms will have important implications for our understanding of human speech, as deficits in self-monitoring and feedback vocal control have been implicated in many human disorders including schizophrenia[29] and stuttering[30,31].

## Methods

**Experimental design.** We recorded neural activity and vocalizations from two adult marmoset monkeys (*Callithrix jacchus*), one male and one female. In order to test feedback-dependent changes in vocal control, vocalizations produced by an animal were altered in real-time and presented to the animal through custom headphones. Neural activity from auditory cortex was recorded using implanted multi-electrode arrays and compared to simultaneously recorded feedback-induced

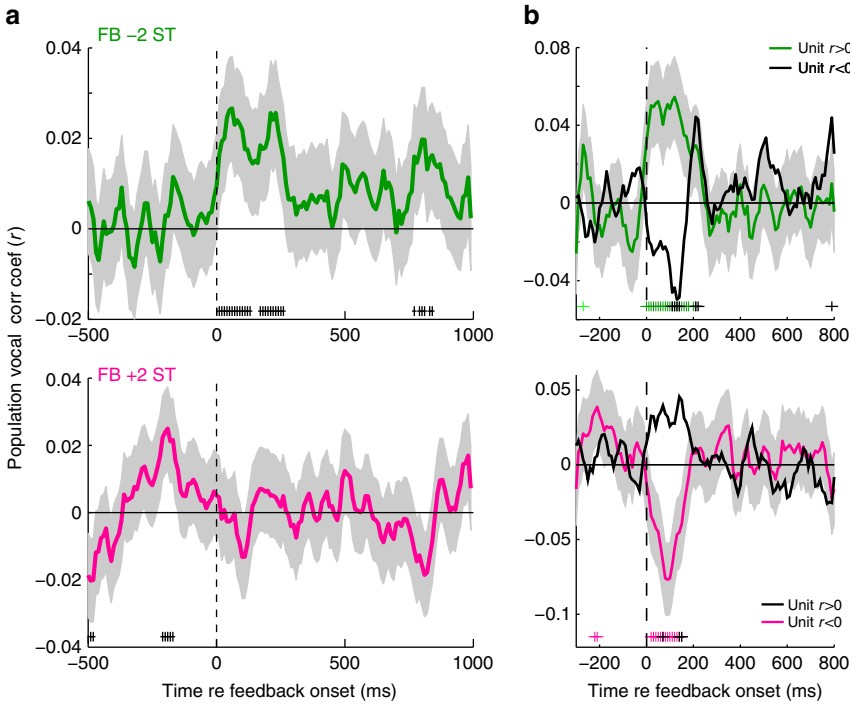

**Fig. 5** Population correlations between neural activity and vocal compensation. **a** Population sliding-window correlation analysis showing timing of correlation coefficients between neural firing in different time bins and vocal compensation after 200 ms. Results are shown separately for −2ST and +2ST (Shaded: 95% CI, +$p < 0.05$). Correlations develop immediately after the start of altered feedback and persist until the time of vocal compensation onset. **b** Population correlations dividing unit into $r > 0$ or $r < 0$, as in Fig. 4. For clarity, error bars have been omitted from the non-compensatory $r$-value units (i.e., $r$-values opposite expected vocal compensation for feedback direction). Significant correlations are indicated (+$p < 0.05$). Sample sizes and statistical details may be found in the Supplementary Note 1

vocal changes. Following several months of neural recordings, we used the same electrode arrays in auditory cortex to deliver focal electrical microstimulation, and recorded the resulting changes in vocal behaviors.

All experiments were conducted under the guidelines and protocols approved by the University of Pennsylvania Institutional Animal Care and Use Committee.

**Vocal recordings**. Following previous methods[16,32], we recorded vocalizations from the marmosets while in their home colony. A subject animal was placed in a small cage with a custom three-walled sound attenuation booth, which improves recording quality but allows free visual and vocal interaction with the rest of the animals in the colony. Recordings were performed with the animal free roaming within the cage, but otherwise unrestrained. Vocalizations were recorded using a directional microphone (Sennheiser ME66) placed ~20 cm in front of the marmoset, amplified (Focusrite OctoPre MkII) and digitized at 48.8 kHz sampling rate (TDT RX-8, Tucker-Davis Technologies, Alachua FL). We used multiple microphones to monitor both the vocalizations produced by the experimental animal as well as sounds from the rest of the colony. We later extracted vocalizations from the recorded signals and classified them into established marmoset call types[33,34] based upon their spectrograms using a semi-automated system. All major call types were produced by animals in this context (phees, triltphees, trills, twitters), however we only examined trill and trillphee vocalizations as only these calls were produced in sufficient numbers to allow comparisons between different conditions.

**Altered (shifted) feedback experiments**. We altered the auditory feedback that a marmoset heard during vocal production in real-time by passing the microphone signal through a commercial effects processor (Eventide Eclipse V4) and modifying the vocal signal to increases or decrease the frequency by ±2 ST for a duration of 1000 ms (Fig. 1a, c). This shift magnitude was chosen based on previous work in both marmosets[16] and humans[4,10,11,18,19]. The sound level of the shifted feedback was calibrated (Crown XLS1000) to ~10 dB sound pressure level (SPL) above the intensity of direct, air-conducted feedback. This increase was necessary to overcome the sound of direct, un-altered feedback of the animal's vocalization. Shifted feedback signals were presented back to the animal through a pair of earbud-style headphones (Sony MDR-EX10LP) modified to attach to the animal's headcap[15]. Typically we only shifted the feedback in one direction (either −2ST or +2ST) in any given recording session. The hardware introduced an approximate 15 ms delay in the shifted feedback signal (for both feedback directions). Previous work using frequency-shifted vocalizations as auditory stimuli found that this feedback delay

did not have a significant affect upon playback auditory responses in marmoset auditory cortex, at least as compared to the effects during vocal production[15].

In order to specifically detect changes in vocal production attributable to shifted feedback, we designed a system to trigger shifted feedback in only a random subset of vocalizations (either 50 or 60% of trials). A controller computer with a National Instruments DAQ card (PCIe-6321) was used to detect vocalizations, and separate these from both background noise and vocalizations from other animals in our colony. A targeted delay of 100 ms between vocal production onset and shifted feedback was used to allow for later within-call comparisons of vocal acoustics. There was variability in the actual triggering time (see Fig. 1b) due to a number of factors, including the presence of background noise in the marmoset colony (delaying vocal detection) and when animals sometimes produced a small 'peep' prior to a trill of trillphee (resulting in early detection and triggering). A smooth onset/offset ramping (10 ms) of the feedback shift was used to minimizing transient frequency discontinuity from the triggering. Both trigger and altered feedback signals were digitized to allow for event timing.

**Implanted electrode arrays and neural recordings**. Both marmosets were implanted with bilateral multi-electrode arrays (Warp 16, Neuralynx, Bozeman MT), one in each auditory cortex. The first animal (m96z) had an array initially placed only in the right hemisphere, which was later removed and replaced at the same time as when we placed an array in the left hemisphere. As a result, the first animal had more units recorded in the right than left hemisphere, while the second animal started with the left hemisphere and had more units on that side. Full details of the electrode array design and recording technique have been previously published[32]. Briefly, the arrays consistent of a 4 × 4 grid of individual moveable sharp microelectrodes (4 MΩ tungsten, FHC, Bowdoinham ME). Neural signals were passed through a unitary-gain headstage (TDT RA16CH) that was attached to the electrode array, and then digitized (TDT System III PZ2 & RZ2). A single electrode on each array was designated as a reference and its signal subtracted from the other electrodes in software to reduce muscle potentials and other movement artifacts. Recordings were performed with the marmoset free-roaming within the recording cage in the colony room, with neural signals relayed by tethers. Neural signals were observed on-line to guide electrode movement and optimize unit quality.

Digitized signals were sorted off-line using custom software and a principle component (PCA) based clustering method, and then classified either as single-unit or multi-unit as previously described[16,32]. Neural signals sampled on different days were treated as individual units even if the electrodes had not been moved between sessions. In addition to single-unit sorting, we also examined multi-unit responses

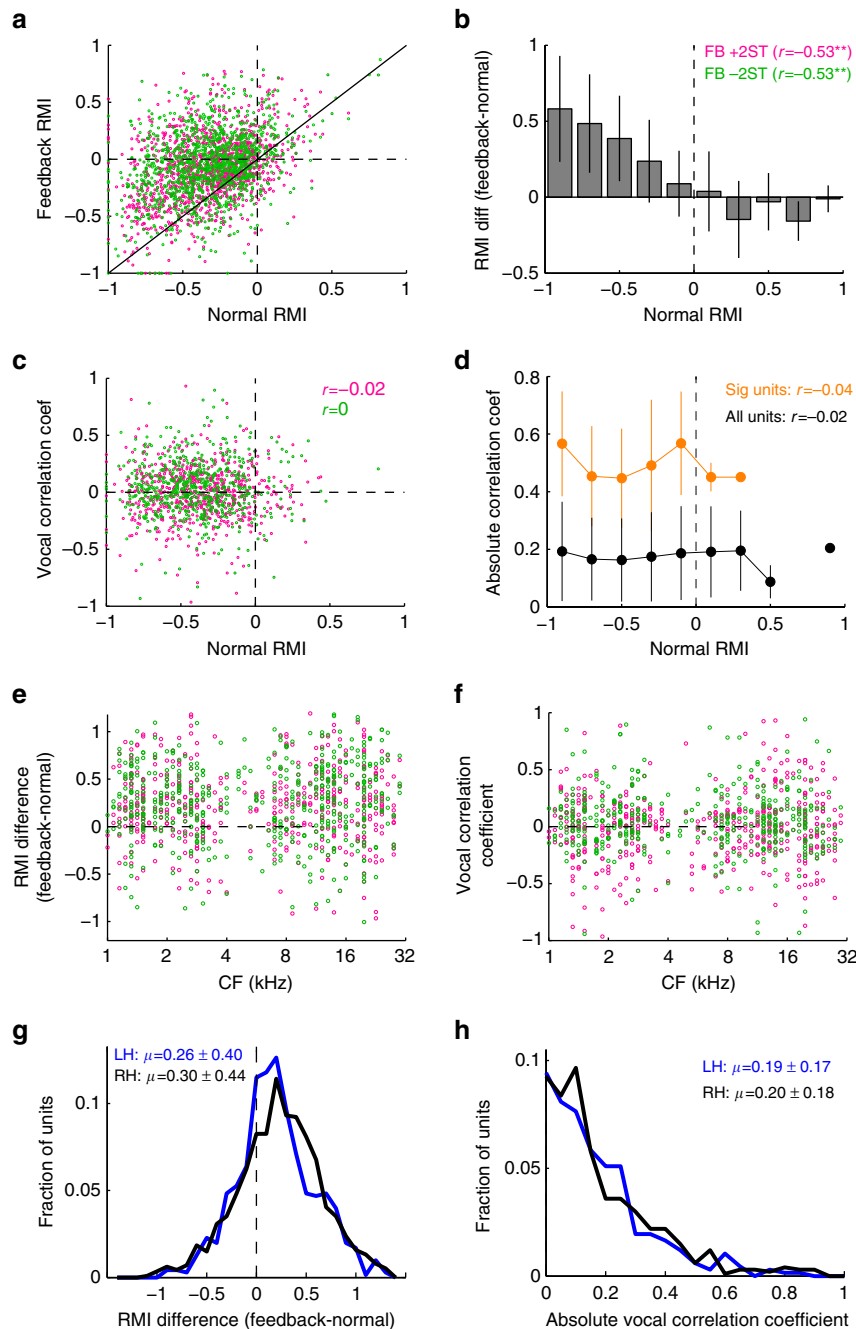

**Fig. 6** Comparison between vocal suppression and compensation predictions. **a** Scatter plot comparing unit vocal responses (RMI) during normal and shifted feedback showing decreased suppression across units. Equal value line (unitary slope) is indicated. FB +2ST evoked an increase in activity compared to normally suppressed vocal responses (RMI change: 0.24 ± 0.31, mean ± std; $p < 0.001$, Wilcoxon signed-rank), as did FB −2ST (0.27 ± 0.37; $p$ < 0.001). **b** Feedback effects plotted as mean Feedback-Normal RMI differences and binned by RMI. Suppressed units (RMI < −0.2) exhibited larger feedback increases than excited units (RMI > 0) which showed more balance effects. This trend of larger activity increases in suppressed units was present for both +2ST ($r = −0.53$, $p < 0.001$) and −2ST ($r = −0.53$, $p < 0.001$) feedback. (Error bars: standard deviation) **c** Scatter plot comparing vocal correlation coefficients and vocal suppression, showing no obvious relationships (+2ST: $r = −0.02$, $p = 0.65$; −2ST $r = 0$, $p = 0.97$). **d** Mean vocal correlation coefficients binned by RMI, shown separately for all data (black) and for only units with significant correlations (orange). **e**, **f** There was also no obvious relationship between frequency tuning (CF) and feedback effects (**e**) or vocal correlation coefficients (**f**). **g**, **h**, Histograms comparing feedback effects (**g**) and vocal correlation coefficients (**h**) between hemispheres (RH: right, LH: left). The right hemisphere showed a slight increase in feedback sensitivity over the left ($p = 0.02$, Wilcoxon rank-sum), but there were no differences for vocal correlation ($p = 0.84$). Sample sizes and statistical details may be found in the Supplementary Note 1

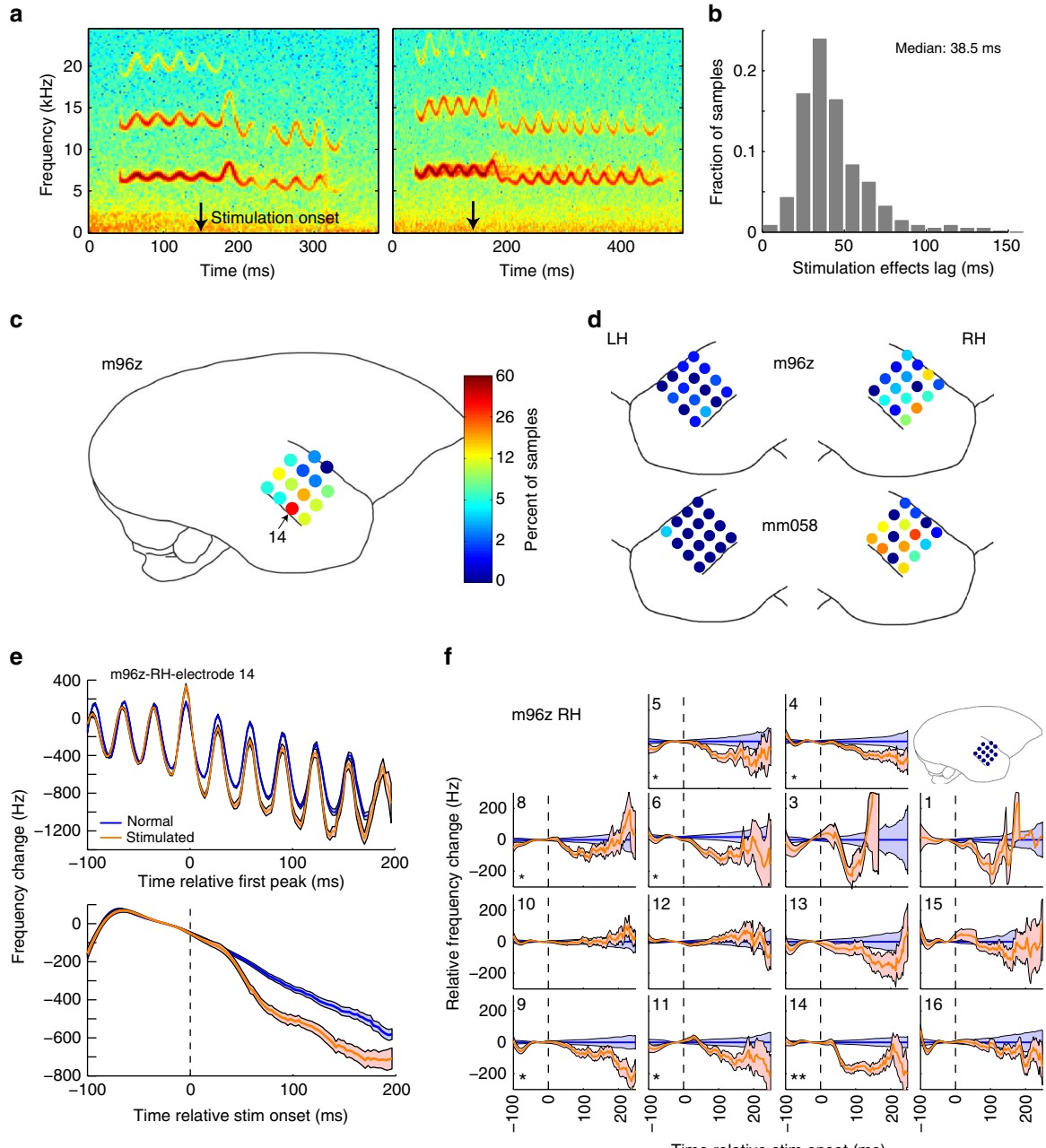

**Fig. 7** Vocal frequency changes during electrical microstimulation of auditory cortex. **a** Two sample trill spectrograms showing abrupt changes in vocal frequencies shortly following stimulation onset (arrows). **b** Distribution of timing latency between stimulation onset and vocal changes demonstrating rapid stimulation effects. **c** Spatial distribution of stimulation effects (percent of trill samples demonstrating frequency changes) in one right hemispheric electrode array. Indicated electrode locations are approximate. **d** Stimulation distribution for two additional left (LH) and right (RH) hemispheric arrays in two animals. Color scale is identical to **c**. **e** Vocal contours for a single stimulation site (indicated in **c**) comparing normal and stimulated mean frequency curves aligned either by the first trill peak after stimulation (top), or smoothed curves aligned by stimulation onset (bottom), demonstrating onset peak and rapid reduction (Error bars: SEM). **f** Vocal frequency contours for a right hemisphere grid (m96z-grid in **c**) demonstrating a mix of electrodes with strong stimulation effects and others with weak to absent changes. (*$p < 0.05$, **$p < 0.001$, Wilcoxon signed-rank)

(MUA) at different recording sites; MUA reflects the summed responses of local populations of neurons[35]. We generated MUA by first subtracting the reference electrode signal from the raw neural signal, then band-pass filtering (300–5000 Hz, 4-pole butterworth), full-wave rectifying, and finally low-pass filtering (500 Hz, 8-pole butterworth) before down-sampling. MUA responses were later used in selected analyses, mainly for comparison to microstimulaton results.

**Auditory stimulus presentation.** Prior to each session of neural recording in the colony, we first characterized the auditory tuning of neurons by the presentation of auditory stimuli. Marmosets were seated in a custom primate chair within a soundproof chamber (Industrial Acoustics, Bronx NY). Auditory stimuli were digitally generated at 97.6 kHz sampling rate and delivered using TDT hardware

(System III) in free-field through a speaker (B&W 686 S2) located ~1 m in front of the animal. Stimuli included tones (1–32 kHz, 10/octave; −10 to 80 dB SPL by 10 dB), bandpass noise (1–32 kHz, 5/octave, 1 octave bandwidth) frequency responses areas (FRA), and wide-band noise stimuli. The center frequency (CF) of a neuron's FRA was determined by the pure tone stimulus with the highest firing rate response, or from bandpass when no tone response was present. We also presented multiple recorded vocalizations at different sound levels, including samples of animals' own vocalizations (previously recorded from those animals) and con-specific vocalization samples (from other animals in the marmoset colony). Vocal stimuli were presented at multiple sound levels, but only those samples overlapping produced vocalization loudness were used for comparisons between vocal production and auditory playback.

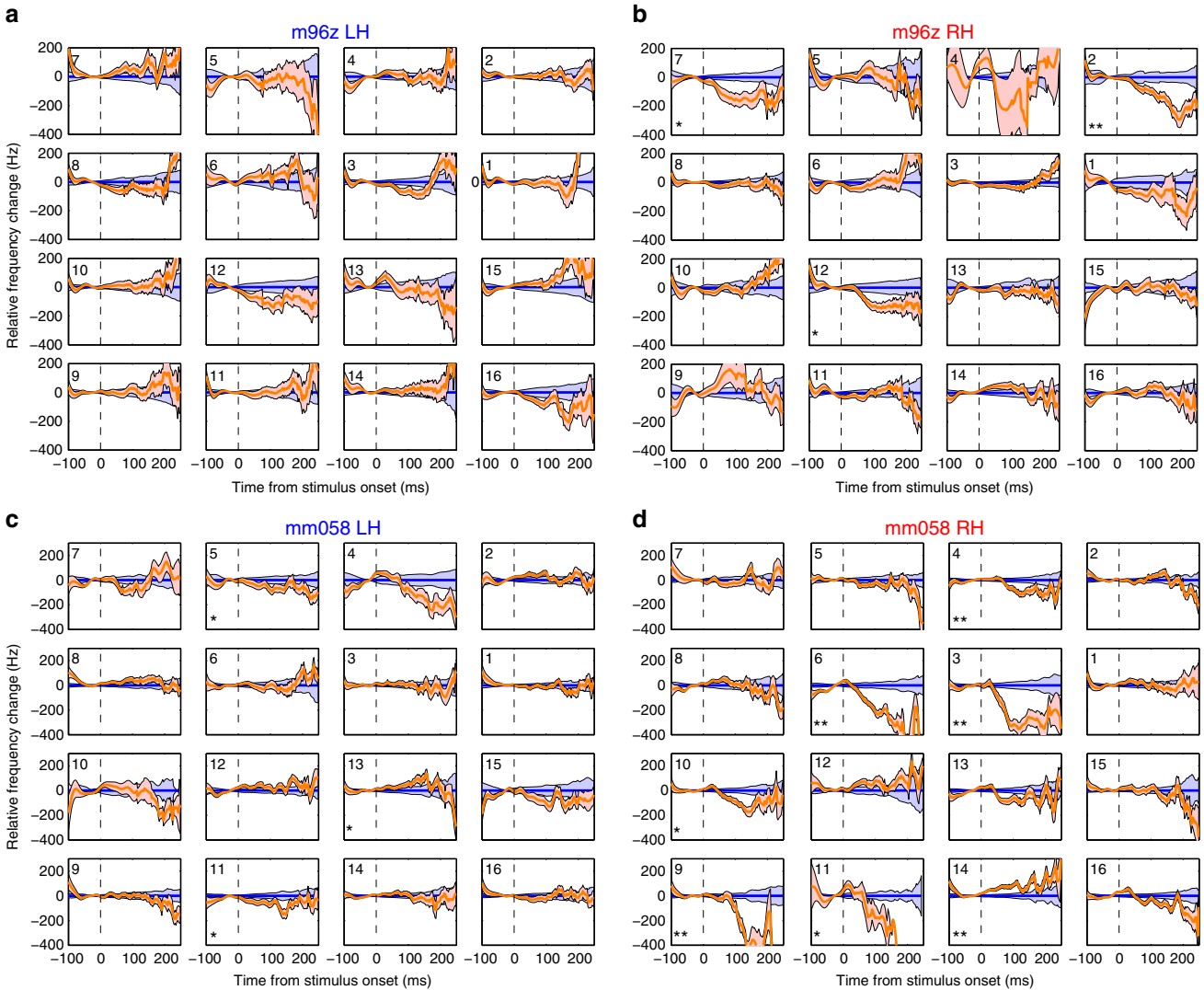

**Fig. 8** Additional examples of microstimulation effects by electrode location. **a** Left and **b** right hemisphere electrode arrays for animal m96z. **c** Left and **d** Right hemisphere electrode arrays for animal mm058. Blue: normal trials; orange: stimulation trials (Error bars: SEM, *$p < 0.05$, **$p < 0.001$, Wilcoxon signed-rank)

All electrodes included units or MUA sites with responses to at least one significant ($p < 0.05$ Wilcoxon rank-sum) auditory stimulus response. Based upon relative responses to tone and noise stimuli, electrodes were judged to likely span both primary (A1) and non-primary (belt, parabelt) auditory cortex[36].

**Recording protocol.** During a typical experimental session, recordings began with auditory stimulus presentation, which was followed by 2–3 h of simultaneous neural and vocal recordings in the marmoset colony. Colony recordings typically started with 30–60 min of baseline (normal) vocal production without shifted feedback, followed by recordings with shifted feedback. During the shifted feedback block, all vocalization feedback was amplified (+10 dB), but only a random subset (50 or 60%) had triggered frequency shifts, as specified above. The direction of frequency shift for a given session was chosen without respect to the frequency tuning of individual neurons studied because we simultaneously recorded multiple different neurons covering the entire hearing range. The order of normal and shifted feedback blocks was changed during some sessions to eliminate temporal interactions. During vocal recordings, animals were given treats at random intervals to maintain a general level of arousal and vocal production.

**Electrical microstimulation.** Following completion of several months of neural recordings, we began electrical microstimulation experiments. Stimulation was triggered to occur ~100 ms following vocal production onset using the identical detection hardware as the feedback experiments, which controlled the current source generator (MultiStim Model 3800 and SIU 3820 isolator, A-M Systems, Carlsborg, WA). Current pulses (100 μA with 0.25 ms biphasic square pulses, 300 Hz, 200 or 1000 ms duration) were delivered on 50% of trials through the recording electrodes. We only tested one electrode per session. Stimulation parameters were

chosen based upon previous experiments in marmoset motor cortex, where stimulation resulted in movement that was presumably a result of cortical activation[37]. Current delivery did not appear to interrupt or prevent ongoing vocal production, and the animals exhibited no abnormal behavior, beyond changes in their vocal acoustic parameters, to indicate a conscious perception. We also tested a smaller current amount (50 μA), but did not note any changes in vocalization (data not shown).

**Vocal data analysis.** In order to determine vocal changes during shifted feedback, we first extracted the fundamental frequency contour for each vocalization (Fig. 1c). Spectrograms were calculated, low-frequency (<2 kHz) background noise was removed, and the frequency with the maximal power in each time bin was calculated to yield a frequency contour. We compared vocal frequency changes due to shifted feedback by comparing average frequencies between the period before the feedback trigger ("pre-feedback") with the period following feedback onset (>200 ms). We only tested vocalizations that continued for at least 250 ms after feedback onset. Reference calls ("normal") were those vocalizations that were amplified but without frequency shifts. The average feedback trigger time (100 ms) was used when calculating expected changes from normal calls. Frequency bins were first averaged within individual calls, and then averaged across calls by feedback condition (normal, +2ST, −2ST). Because of considerable day-to-day variability of vocalizations, each session's mean change (feedback period—pre) in normal calls was subtracted from a given session's individual calls prior to averaging.

An additional analysis was performed to determine the timing of feedback compensation by averaging frequency contours during each feedback condition. Raw frequency contours were first smoothed to remove oscillations from the trill

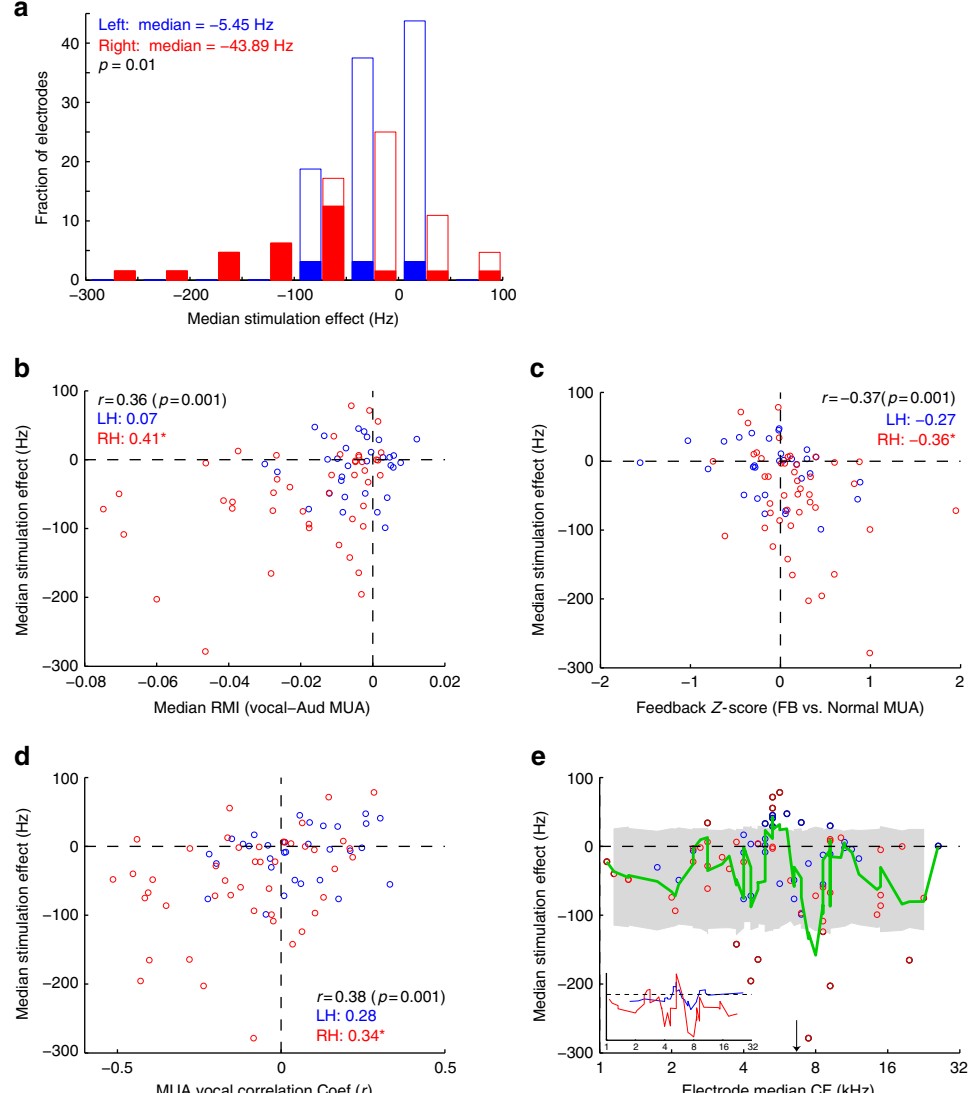

**Fig. 9** Correlation between neural recording and microstimulation effects. **a** Histogram of median vocal frequency changes during microstimulation for each electrode site (Shaded: $p < 0.05$, Wilcoxon signed-rank within electrode). Results are plotted separately for right and left hemispheric electrodes, with stronger effects on the right ($p = 0.01$, Wilcoxon rank-sum between hemispheres). **b** Scatter plot comparing stimulation effects and the degree of vocal suppression (vocal-auditory RMI) of multi-unit activity (MUA) at the matched recording site. Significant correlations were present for the population, as well as in the right hemisphere (*$p < 0.01$), but not left hemisphere. **c** Scatter plot showing negative correlation between stimulation and feedback sensitivity. **d** Plot comparing microstimulation and vocal compensation correlation coefficients. Sites with negative neural-vocal correlations showed greater decreases in vocal frequency during stimulation, with positive correlation sites showing the opposite. **e** Comparison of microstimulation effects by electrode frequency tuning, demonstrating largest decreases clustered around 8 kHz, just above mean trill frequency (arrow), and increases around 5–6 kHz, just below mean trill frequency (Green: moving average; shaded: 95% CI). Inset: moving average curves separated by right (red) and left hemispheres (blue)

calls (i.e., Fig. 1c) by low pass filtering, followed by subtraction of the mean frequency in a window around the feedback trigger (−50 to +50 ms). Frequency contours were then aligned by feedback onset, using 100 ms for the reference normal (amplified only) calls. The second animal (mm058) had a tendency to preface its vocalizations with a short "peep" sound, which caused multiple early shifted feedback triggers, as a result we only included the 50 ms before the trigger in this animal. Also, because of this early triggering, significant variability was noted in the averaged pre-feedback frequency contour of this animal, a result of variable timing of the vocal onset. This was corrected by calculating a linear scaling factor to better normalize the pre-trigger frequency contours between feedback and normal conditions, which was then applied to all subsequent time points. To remove expected frequency contours, and reduce the effects of day to day variability, as before, the mean contour of normal calls for each session was subtracted from that session's individual call contours prior to further averaging. Final results showed increasing standard error (SEM) in later portions of the calls due to both increasing variability between different calls and decreasing number of samples. We determined the onset timing of significant feedback compensation in the mean frequency contours by calculating $p$-values for each individual time bin (2.6 ms)

using Kruskal–Wallis non-parametric ANOVAs. False discovery rate (FDR) corrections were applied for the multiple time bins tested, with $p < 0.001$ considered significant. Due to the large number of calls compared, possibly resulting in additional falsely positive $p$-values, we also required mean feedback responses deviate from normal by 1.96 SEMs (95% CI for the mean estimate). For individual analysis of the +2 and −2 responses within individual time-bins, post hoc Bonferroni corrections were applied, followed by FDR corrections across time bins.

**Neural data analysis.** Responses to individual vocalizations were calculated by comparing the firing rates before and during self-initiated vocalizations. Pre-vocal activity was assessed by randomly selecting 1000 time bins, with lengths matching vocal duration, from the 4 s preceding vocal onset, and calculating the average firing rate of 500 bins with the quietest microphone signal (acoustic background), thereby minimizing the effects of colony noises on calculations. A window of 500 ms immediately before vocal onset was excluded from this calculation because of previous work[14] indicating pre-vocal suppression. The response to each

vocalization was quantified using a normalized rate metric, the vocal Response Modulation Index (RMI), defined as:

$$RMI = \left(R_{vocal} - R_{prevocal}\right) / \left(R_{vocal} + R_{prevocal}\right),$$

where $R_{vocal}$ is the firing rate during vocalization and $R_{prevocal}$ is the average rate before vocalization. An RMI of −1 indicated complete suppression of neural activity during vocalization and +1 indicated strongly driven vocalization responses, a low pre-vocal firing rate, or both. Vocalization responses that failed to elicit at least three spikes before or during the vocal period were excluded from analysis. The overall response of a neuron to vocalizations was assessed by averaging the RMI from multiple vocalization responses.

The effect of shifted feedback on auditory cortex neurons was determined by calculating RMIs for vocalization responses under both normal (unamplified) and shifted feedback conditions, and comparing the average RMI from both conditions. Due to potential small changes in pre-vocal background activity from feedback amplification, the mean pre-vocal firing during shifted feedback trials was corrected by subtracting the difference between normal and feedback mean pre-vocal firing rates. The RMI difference between shifted feedback and normal vocalizations was used to quantify feedback responses, with positive differences indicating increased neural activity during frequency shifts. The effects of amplified feedback alone were examined in a subset of data, but were found to be small in most neurons and were not subjected to further analyses.

Comparisons of feedback effects on suppressed (RMI < −0.2) neural populations were also made by calculating peri-stimulus time histograms (PSTHs). PSTHs were calculated by averaging neural responses to vocalizations (10 ms bins), aligned by the onset of each vocalization, and then subtracting the spontaneous pre-vocal activity. PSTH calculations for individual neurons, for display purposes only, used a 20 ms bins followed by smoothing. Population PSTHs were calculated by averaging spontaneous-subtracted PSTHs of individual units, but were not otherwise normalized.

The relationship between neural activity and vocal compensation was examined by comparing neural activity during the early period of shifted feedback (firing rate during 0–200 ms after feedback onset) to subsequent vocal compensation in the compensation period (>200 ms). The degree of vocal compensation for individual trials was calculated by z-scoring the vocal frequency change relative to the mean normal frequency change as above. We compared PSTH curves for both single-unit/MUAs, as well as single-unit population averages, between calls with z-scores > 0 vs. <0, corresponding to "compensated" and "non-compensated calls". A z-score > 0 (increased frequency) would be considered "compensated" for FB −2ST, whereas a z < 0 (decreased) would be considered "compensated" for FB +2ST, given the behavioral compensation directions demonstrated in Fig. 1.

We further analyzed the relationship between vocal compensation (z-score) and neural activities (firing rate) by calculating correlation coefficients between early-feedback (0–200 ms) firing rates and subsequent vocal compensation (>200 ms) for each individual unit. Only units with at least five vocalizations of a given call-type were included when calculating such correlations, though many units had far larger numbers of vocal samples. The first animal (m96z) tended to make more vocalizations per session than the second, and therefore had more units available for correlation analysis. Timing of this correlation was also examined by calculating correlation coefficients using a sliding time window (10 ms) of neural activities, after smoothing (five point moving average), between firing rates and z-scored vocal frequency compensation (fixed time window, >200 ms). Results were calculated both for individual units and at the population level.

**Microstimulation data analysis**. The effects of microstimulation on trill calls were determined by calculating frequency contours, both with and without removal of trill oscillations. Analysis of the timing of stimulation effects was performed by examination of vocal frequency contours by a reviewer, blinded to the onset time and presence of stimulation for a trial, to determine the presence or absence of abrupt changes in the expected oscillatory pattern. Onset of stimulation effects was determined by the first visible abrupt change, either to an abnormal peak or a sudden decrease in the trilling contour. The percent of vocalizations with these effects was calculated for each electrode. Subsequently, quantitative analysis was performed to verify these changes by comparing mean frequency contours for stimulation and normal trials. These were performed by separately aligning both to the stimulation onset and to the first trill frequency peak following stimulation onset, which better revealed early abrupt changes. When comparing across electrodes, normal (no-stimulation) vocal frequency contours were subtracted from stimulation trial contours as was done for vocal compensation behavior.

Additional metrics of stimulation effects were used to compare stimulation results to those from neural recordings. The median change in vocal frequency, compared to normal calls from the same session, was calculated for individual electrodes. To measure the stable stimulation response, and exclude variable onset latency, only the period of 50–200 ms after stimulation onset was included. For each stimulation trial, the median frequency change from normal was calculated over this time interval. Composite results for individual electrodes were calculated using median values over multiple stimulation trials from a given session. Median was chosen for this analysis due to the small numbers of samples, and to reduce the effects of outliers. Neural recordings were selected as responses from the most

recent recording session prior to stimulation for a given electrode. Because microstimulation likely affects a local population of neurons, rather than individual units, MUA analysis was chosen as the comparison rather than the single-unit data. When comparing MUA and frequency tuning, we ordered stimulation responses by median columnar CF, and performed a 3-point moving average to create a stimulation "tuning curve". We also calculated a shuffled-corrected confidence interval for this comparison by randomizing the CF-stimulation pairs 1000 times, and measuring the intervals containing 95% of the shuffled results.

**Statistical methods**. Unless otherwise noted, all statistical tests on neural data were performed using non-parametric methods. Wilcoxon rank-sum and signed-rank tests (two-sided) were used to test the differences between matched and un-matched distribution medians, respectively. Kruskal–Wallis ANOVAs, with Bonferroni corrections for multiple comparisons, were used when comparing more than two conditions for behavioral or neural results. Behavioral data were compared with traditional ANOVAs due to large sample sizes. Correlation values within individual unit, and between unit parameters were calculated with Pearson correlation coefficients, with p-values and confidence intervals calculated from the t-distribution, and corrected for multiple-comparisons. Correlations between stimulation and neural parameters were calculated with Spearman rank correlations, due to smaller sample size, with permutation test verification of significance. For moving-window methods of measuring time changes, p-values were first calculated for individual time bins, and then FDR corrected for multiple time-point comparisons. Multivariate linear regressions were performed to disentangle possible correlated variables in predicting stimulation effects. p-values < 0.05 were considered statistically significant throughout.

**Data Availability**. The data and computer code uses to support the findings of this study are available from the corresponding author upon request.

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

## Acknowledgements

This work was supported by NIH grant DC014299 (SJE), and funding from the Trio-logical Society Clinician-Scientist Development Award. We thank T. Coleman and P. Sayde for assistance in animal training and care, and Y. Cohen for comments on this manuscript.

## Author contributions

S.J.E. and J.T. contributed equally to designing and conducting experiments, data analysis, creation of figures, and manuscript preparation.

## Additional information

**Competing interests:** The authors declare no competing interests.

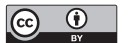

