## [Peer Review File · Nature Communications]

Reviewers' comments:

Reviewer #1 (Remarks to the Author):

The authors' group has reputation of studying auditory cortical activity during vocalization, or auditory version of efference copy, cortical suppression, related to vocalization in marmoset auditory cortex. It is an interesting and unique theme, and would appeal to board readers' interest in how an animal and its auditory cortical neurons monitor vocalization of itself.

This study stepped further into addressing the relationship between changes in neuronal activity and vocalization feedback, by manipulating the feedback sound direction and timing. By introducing delayed feedback of frequency shifted self-vocalizing sound, they found modulated neuronal and behavioral events in the order of rebound from neuronal suppression followed by alteration of the pitch of ongoing vocalization. When feedback sound did not induce the rebound during some vocalization trials, then the following modulation of vocal pitch did not happen either. Furthermore, when delayed electrical stimulation was administered during vocalization to rebound the vocalization-suppressed neuronal activity, pitch modulation occurred. These results taken together suggest that auditory cortical activity during vocalization acted not only to monitor vocal errors at least in terms of pitch, but also to drive behavioral corrections.

Even though the data variability looks large, it would be fair to consider the technical difficulty of the experiments done in this study. Part of reason probably was that the animals' behavior under investigation was not controlled by experimenters, but rather initiated voluntarily by animals themselves. But to study vocalizations, it was rather advantageous as the experiments were conducted during natural behavioral conditions. I think the findings are appealing and provide supportive experimental evidences for models of auditory cortex functions during vocal behaviors, which would be of interest of many readers.

There were several ambiguities in data analyses that deserve more clarity, and questions to which I would appreciate some answers.

- (1) The study shows possible hemispheric biases of stimulation effects. How about similar hemispheric biases in effects of feedback sound or neuronal correlation with vocal modulation following feedback? Also, topographic patterns like medial-lateral gradients?
- (2) Previous study from same authors showed that vocal suppression started earlier than vocalization. However, such early onset of suppression is not clear to me in figures of the manuscript. Did the suppression start before vocalization onset in the dataset of this study?
- (3) As described in Discussion, circuitry between auditory and motor command is an open question. It would be nice if authors could give more insights into the issue. For example, I guess no anatomical studies showed direct connection between auditory and motor cortex, but connections of auditory cortex with other parts of brain.
- (4) When neuronal units had negative or positive correlation with frequency shift of vocalization for one call type, do they exhibit same signs of correlation for other call types?
- (5) Is it possible that any two different populations of units (e.g. vocalization-suppressed and vocalization-excited, positive and negative correlation with vocal changes) differed in spike waveforms? I'm wondering neuron types may be different, as another possibility in

addition to the possibility of push-pull described in the manuscript.

(6) Figure 1B. The onset of feedback fluctuated around 100 ms. I guess there were technical reasons, like difference in onset detection between the sound monitoring and offline acoustic analyses, that many readers would be not familiar with. So, it would be nice if authors briefly clarify why.

(7) According to Fig.1B, the feedback sound started about 90 ms on average after the vocalization onset, and 200 ms as in Fig.1C was the longest delay. Isn't it better to use more typical delay time as an example to show? Also, according to the text and Fig.1E, pitch change of vocalized sound started around 200 ms after the onset of feedback. That would be at 500 ms for the example of +2ST shown in Fig.1C. However, it is difficult to see a downward change in the pitch at 500 ms, particularly compared to strong effect of electrical stimulation seen in Fig.7. Is that because of small change (50 Hz) compared to the spectral range of the figure? Since -2ST produced larger changes in pitch, according to Fig.1D, Isn't it better to use an example of -2ST?

The figure appears to have superimposition of additional harmonics in the Feedback period. Is that an envelope of feedback sound leaked from earbuds?

During the Feedback, envelope phases of frequency-shifted (=feedback) and non-shifted (=vocalized) sounds do not align with each other. I guess that is presumably due to the sound monitor-feedback system. Did such phase changes occur systematically along with pitch modulation in the feedback sounds? There is a chief possibility that any neuronal effects attributed to the pitch modulation could be caused by the altered envelope phase.

(8) Figure 1E indicates Kruskal-Wallis test. I assume that significant difference could be either -2ST or +2ST. Can it be replaced by post-hoc comparisons to indicate which of +2ST and -2ST were significant? It would be nice to know the onset of effects separately for 2 types of feedback.

(9) Figure 2A. What does the shaded area in the figure indicate? Is it the duration of vocalization, like same authors' previous publication? One thing I notice in the figure is that the duration of vocalization during shifted feedback may be shorter. Is that so across dataset?

(10) Figure 4B. 8.8% for -2St and 8.8% for +2ST. Do those 8.8% units form functional clusters or scatter over cortices?

Also, it may be better to indicate the range of the number of vocalization call trials used to calculate the correlation coefficients, or otherwise to limit the analyses to units/call types for which the number of call trials are large enough to examine correlation.

Were there no bias between 2 animals, e.g. like units of one animal showed only negative correlation and units of another showed only positive? Even if there were such a bias, it should not affect the results of effects of delayed shifted feedback sounds, but it would be nice to know.

It may be better to tabulate them to show the number of units instead of percentages, with sorting of animal/hemisphere/change direction, etc. I guess that would also tell the number of units constituting the data for Figure 5.

(11) Even though the percentages of units correlated with behavioral adjustments were small, all those units included in the analyses, even non-correlated ones, exhibited vocal suppression/enhancement. Correct?

And, were all those units, regardless of behavioral correlation significant or not, used for analyses in Figure 6? How about limiting the data samples to those 8.8 % units with

significant behavioral correlation for the analyses shown in Figure 6C? The question I ask here is whether behaviorally correlated units are limited to strong vocal suppression or not.

(12) Cortical electrical stimulation. A note about the intensity of stimulus current is described in Methods. How about the structure of stimulus pulse train? I guess it was planned to mimic rebound neuronal activity during feedback somehow. Whatever the reason was, it may better be described.

(13) Ordinate axes label: "Firing Rate (spk/s)" in figures. PSTH in figures intuitively show inhibitory responses. However, those PSTH are not exactly plotting the firing rate, but the changes in the firing rate from pre-feedback. It may make sense to re-label them as "firing rate change", which would be similar to "Frequency Change (Hz)" in Fig 1.

(14) A description in Results: "both the single-unit (Fig. 2a) and population (Fig. 2b) levels" should be "both the single-unit (Fig. 2b) and population (Fig. 2c) levels"

(15) Figure 3D was derived by sliding window of correlation. Was the sliding applied only to the firing rate while vocal compensation was fixed at >200 ms, or was the sliding window applied to both firing rate and vocal compensation similarly somehow? It is not clear from the descriptions in both the figure legend and methods.

(16) Figure 4. Some labels on top of figures are bold, some are italic. Does it mean anything?

(17) "mean+std" in Figure legends. Does "std" mean SEM or SD?

(18) Fig.9a shows median changes in vocal frequency. I guess median values were derived from changes of individual vocalizations. Methods describe that "the period of 50-200 ms" was used. Please clarify that estimated values were the mean change during this period, or else.

(19) "the most commonly observe stimulation effect" -> "observed"

(20) "was performed by manually examination by a blinded reviewer" -> ?

(21) "begasan" -> "began"

Reviewer #2 (Remarks to the Author):

The submitted manuscript, 'Auditory cortical activity drives feedback-dependent vocal control in marmosets' examines auditory/feedback sensory-motor processing in marmosets. The authors demonstrate that like humans, marmosets alter their vocalizations in the presence of altered feedback in a compensatory manner. Furthermore, they demonstrate that a sub-set of auditory cortex neurons alter their firing rates during this altered feedback. Finally, the authors show that electrical stimulation (mostly in the RH) results in altered vocalizations, and that the degree firing/site changes during compensation correlates with the amount of acoustic change produced by the stimulation.

Overall, I find this manuscript to be a novel and interesting addition to the literature. The combination of behavioral, neural, and stimulation results is a particular strength. I have a few concerns however.

1.The authors do not reference figure 2c in the text.

2. Fig 3c: a fit would be helpful for this figure.

3. Fig 4: How are the firing rates normalized here? This should be explained in the caption and I assume your Y axis Label should then change.

4. For the 8.8% of units that displayed significant correlation coefficients: did you correct for multiple comparisons? If not, please do.

5. "These effects were stronger for units whose correlation was in the same direction as the expected vocal compensation" Please quantify.

6. Figure 7-8: I think the individual plots can be moved to a supplement.

7. Figure 7-8: it would be useful to summarize these plots into some kind of statistical analysis.

We would like to thank you and the two reviewers for the opportunity to revise our manuscript. Overall, the two reviews were very positive in their tone, both in terms of the results as well as the interest to a broad audience. We found their questions very useful and insightful, focused more on some of the details and secondary questions rather than the main message, a message they appeared to appreciate as strong and interesting. We have performed additional analysis and revised the manuscript based upon their questions and comments. Below can be found a copy of the reviewers' questions (in italics) as well as our responses, and citations to the location of changes in the manuscript. Changes made to the manuscript have been colored in red to aide the reviewers in their re-evaluations.

Reviewer 1

(1) The study shows possible hemispheric biases of stimulation effects. How about similar hemispheric biases in effects of feedback sound or neuronal correlation with vocal modulation following feedback? Also, topographic patterns like medial-lateral gradients?

This is an excellent question, one which we had examined previously but excluded for brevity. We have now included this in the revised manuscript. There was a weak but significantly stronger feedback effects in the right hemisphere ($p=0.016$), but no differences for vocal correlations ($p=0.84$). We have included this as Figures 6 f-g, and discussed both on page 7 and 8. We also examined the effects of medial-lateral gradients, but there were no consistent patterns ($p>0.05$ for both feedback and vocal correlation). This is also discussed on page 8.

(2) Previous study from same authors showed that vocal suppression started earlier than vocalization. However, such early onset of suppression is not clear to me in figures of the manuscript. Did the suppression start before vocalization onset in the dataset of this study?

The reviewer astutely notes that pre-vocal suppression is not as obvious in the example neurons shown in this manuscript as in some of our past work. Close examination of figure 2B shows that suppression did begin pre-vocally for this unit, and there is weak trend in the population data in Fig 2C. This is an interesting issue that has come up before, as we did not note as strong pre-vocal suppression in our 2008 paper, but did see it in the 2013 analysis. The difference stems from whether or not all responses from a unit are averaged, or only those with strong suppression. If all responses are averaged, the pre-vocal effect sometimes "washes out." We suspect the pre-vocal suppression correlations with the total suppression, which varies from call to call, even without altered feedback. This may imply something about the efferent copy signal, which we hope to investigate more fully in future papers, but is beyond the scope of this manuscript.

(3) As described in Discussion, circuitry between auditory and motor command is an open question. It would be nice if authors could give more insights into the issue. For example, I guess no anatomical studies showed direct connection between auditory and motor cortex, but connections of auditory cortex with other parts of brain.

We thank the reviewer for this suggestion. We have added additional discussion and references about possible pre-motor connections via the arcuate fasciculus (page 11), although the functional role of such connections remains uncertain.

(4) When neuronal units had negative or positive correlation with frequency shift of vocalization for one call type, do they exhibit same signs of correlation for other call types?

This is an excellent suggestion. We compared the correlation coefficients between trill and trillphee vocalizations for the same unit and found an overall correlation (a correlation of correlations) of $r=0.006$ ($p=0.89$), suggesting little relationship. We now mention this finding on page 6. This is, perhaps not surprising, as we have previously seen that units often have diverse responses to the different vocalizations (see 2013 paper). To what extent this reflects differing vocal acoustics (trillphees tend to be slightly higher frequency) is one interesting possibility, but we do not have sufficient data for each unit to evaluate further.

(5) Is it possible that any two different populations of units (e.g. vocalization-suppressed and vocalization-excited, positive and negative correlation with vocal changes) differed in spike waveforms? I'm wondering neuron types may be different, as another possibility in addition to the possibility of push-pull described in the manuscript.

This is certainly an intriguing hypothesis, and one we have looked at repeatedly over the years (i.e. FS interneurons as excitatory cells, RS pyramidal cells as inhibited ones). Unfortunately no such pattern has ever materialized, probably similar to the lack of FS/RS differences for simple/complex cells in V1. We re-examined the current data (shown right), but did not see any regular patterns. Red is FS (<0.35 ms peak-trough time) vs blue RS. Aside from some non-significant differences in RMI (excitation vs inhibition), everything else is nearly identical.

(6) Figure 1B. The onset of feedback fluctuated around 100 ms. I guess there were technical reasons, like difference in onset detection between the sound monitoring and offline acoustic analyses, that many readers would be not familiar with. So, it would be nice if authors briefly clarify why.

Thank you for the suggestion, we have clarified the issue (page 16) to discuss causes of variability including colony noises as well as occasional pre-triggering by small peeps.

(7) According to Fig.1B, the feedback sound started about 90 ms on average after the vocalization onset, and 200 ms as in Fig.1C was the longest delay. Isn't it better to use more typical delay time as an example to show? Also, according to the text and Fig.1E, pitch change of vocalized sound started around 200 ms after the onset of feedback. That would be at 500 ms for the example of +2ST shown in Fig.1C. However, it is difficult to see a downward change in the pitch at 500 ms, particularly compared to strong effect of electrical stimulation seen in Fig.7. Is that because of small change (50

Hz) compared to the spectral range of the figure? Since -2ST produced larger changes in pitch, according to Fig.1D, Isn't it better to use an example of -2ST?

Thank you for the suggestion, we have substituted a new example Fig1B with a more typical onset of the feedback. As to the question about difficulty in visualizing trends, the reviewer is correct. The overall changes are on the order of 100 Hz at peak (30-50 average), which is difficult to visualize given the spectral range of the call. It is also difficult to evaluate on individual trials due to considerable variability with marmoset vocalizations, hence the need for the analysis in Fig 1E.

The figure appears to have superimposition of additional harmonics in the Feedback period. Is that an envelope of feedback sound leaked from earbuds?

The reviewer is correct. We now point this out in the figure legend (page 30) for readers.

During the Feedback, envelope phases of frequency-shifted (=feedback) and non-shifted (=vocalized) sounds do not align with each other. I guess that is presumably due to the sound monitor-feedback system. Did such phase changes occur systematically along with pitch modulation in the feedback sounds? There is a chief possibility that any neuronal effects attributed to the pitch modulation could be caused by the altered envelope phase.

There is an approximate 15ms delay in the feedback as a result of the vocal processor. This did not change between +2 and -2 semi-tones, and we did not test other shift magnitudes for this experiment. We now note this on page 15. We addressed the possibly contribution of this delay, albeit indirectly, in our 2008 paper where we used more extensive auditory (playback) controls. Those playback experiments included the same phase delay as seen in the shifted stimuli, and found relatively weak effects on sensory responses, especially as compared to the strong effects during vocal production. This is also now discussed on page 15. Obviously, this could also be addressed by studying neural responses to both +2 and -2 in the same neurons, since both have similar phase delays. We have done this in a small number of units, and found diverse effects, but with most neurons showing different responses between +2 and -2 conditions. We are presently doing more extensive recordings to do such multi-shift comparisons for individual neurons, with a goal of looking at "tuning" of feedback sensitivity.

(8) Figure 1E indicates Kruskal-Wallis test. I assume that significant difference could be either -2ST or +2ST. Can it be replaced by post-hoc comparisons to indicate which of +2ST and -2ST were significant? It would be nice to know the onset of effects separately for 2 types of feedback.

The onset of compensation was 135ms for +2ST, and 244 for -2ST. We now note this in the text (page 4).

(9) Figure 2A. What does the shaded area in the figure indicate? Is it the duration of vocalization, like same authors' previous publication? One thing I notice in the figure is that the duration of vocalization during shifted feedback may be shorter. Is that so across dataset?

Thank you for catching the omission. The shaded area represents the duration of vocalization we now note this in the figure legend (page 30). There was no systematic differences between vocal durations during altered feedback and non-feedback, in fact the altered feedback vocalizations were slightly longer (an average difference of 8ms, now mentioned on page 4).

(10) Figure 4B. 8.8% for -2St and 8.8% for +2ST. Do those 8.8% units form functional clusters or scatter over cortices?

This is a very interesting question. There as no systematic organization of the correlation coefficients or for the significantly correlated units (nor was there hemispheric differences, which can be inferred from the revised figure 6). We also suspect the course sampling of the electrode array (>500 microns between electrodes) precluded any detection of local clustering. We now note this absence in the manuscript (page 6).

Also, it may be better to indicate the range of the number of vocalization call trials used to calculate the correlation coefficients, or otherwise to limit the analyses to units/call types for which the number of call trials are large enough to examine correlation.

We required a minimum of 5 vocalizations in a feedback condition in order to calculate correlations, though the full range was sometimes as high as 126. The number of vocal trials in a given session was, obviously, something beyond our control, and some sessions/units with lower call numbers may have artificially reduced the statistical significance for some units. We now note these minimal vocal numbers in the methods (page 23).

Were there no bias between 2 animals, e.g. like units of one animal showed only negative correlation and units of another showed only positive? Even if there were such a bias, it should not affect the results of effects of delayed shifted feedback sounds, but it would be nice to know.

There we no significant differences between the two animals, m96z had a mean correlation of 0.015+/-0.26, mm058 was 0.007+/-0.26 (p=0.46). Nor were the differences for feedback sensitivity (0.28+/-0.41 vs 0.29+/-0.45, p=0.91). We now note the absence of animal differences on page 7.

It may be better to tabulate them to show the number of units instead of percentages, with sorting of animal/hemisphere/change direction, etc. I guess that would also tell the number of units constituting the data for Figure 5.

We thank the reviewer for the suggestion, and have provided the table for the revised manuscript, cited on page 6. We also note causes for different unit numbers per animal in the methods section (page 23), which was partially due to the number of vocalizations made per session.

(11) Even though the percentages of units correlated with behavioral adjustments were small, all those units included in the analyses, even non-correlated ones, exhibited vocal suppression/enhancement. Correct?

And, were all those units, regardless of behavioral correlation significant or not, used for analyses in Figure 6? How about limiting the data samples to those 8.8 % units with significant behavioral correlation for the analyses shown in Figure 6C? The question I ask here is whether behaviorally correlated units are limited to strong vocal suppression or not.

The reviewer is correct, all units were included in analyses, even those that were not significantly correlated, and both correlated and non correlated units had vocal suppression, as can be seen in Figure 6C. The reviewer's suggestion for an additional plot is an excellent one, and we now plot a new Figure 6D (legend page 33) showing the strength of vocal correlation (absolute value) vs RMI, both for all data and only for significantly correlated units. We binned results as in Fig 6B to make it easier to visualize. There was a trend towards stronger correlation with stronger suppression, but this was not significant ($r=-0.02$, $p=0.41$ all units; $r=-0.03$ $p=0.77$ significant units). However it is hard to interpret the relative role of suppression given a limited number of units with strong vocal excitation.

(12) Cortical electrical stimulation. A note about the intensity of stimulus current is described in Methods. How about the structure of stimulus pulse train? I guess it was planned to mimic rebound neuronal activity during feedback somehow. Whatever the reason was, it may better be described.

We give the details in the Methods on page 19. They were 100 μ A with 0.25ms biphasic square pulses, 300 Hz, 200 or 1000 ms duration. These parameters were chosen from the literature, from the (only!) paper using microstimulation in marmoset cortex. That paper was in motor cortex, and showed stimulation-induced movement, presumably reflecting cortical activation. We now discuss this choice on page 19.

(13) Ordinate axes label: "Firing Rate (spk/s)" in figures. PSTH in figures intuitively show inhibitory responses. However, those PSTH are not exactly plotting the firing rate, but the changes in the firing rate from pre-feedback. It may make sense to re-label them as "firing rate change", which would be similar to "Frequency Change (Hz)" in Fig 1.

An excellent suggestion, thank you.. We have made the requested changes where applicable (Figures 2-4). We also note PSTHs as spontaneous subtracted in methods (page 23).

(14) A description in Results: "both the single-unit (Fig. 2a) and population (Fig. 2b) levels" should be "both the single-unit (Fig. 2b) and population (Fig. 2c) levels"

Thank you for catching that, fixed.

(15) Figure 3D was derived by sliding window of correlation. Was the sliding applied only to the firing rate while vocal compensation was fixed at >200 ms, or was the sliding window applied to both firing rate and vocal compensation similarly somehow? It is not clear from the descriptions in both the figure legend and methods.

Thank you. The firing rate was sliding window, but the vocal compensation was fixed for >200ms. This has been clarified in the Methods (page 23).

(16) *Figure 4. Some labels on top of figures are bold, some are italic. Does it mean anything?*

The bold was meant to clarify those units where the correlation was in the same direction as the appropriate compensation. Italic was those counter-direction. We have removed the italics, and clarified the bolding in the figure legend (page 31).

(17) “mean+std” in Figure legends. Does “std” mean SEM or SD?

std is the standard deviation. This has been clarified in the first use for the Fig 4 legend.

(18) *Fig.9a shows median changes in vocal frequency. I guess median values were derived from changes of individual vocalizations. Methods describe that “the period of 50-200 ms” was used. Please clarify that estimated values were the mean change during this period, or else.*

For each stimulation trial, we first calculated the difference from the normal vocal frequencies for that same session, then took the median over the 50-200ms interval (chosen to exclude the onset effects and only measure the steady state stimulation effect). We then took the median of these individual call values to summarize the stimulation effect for a given electrode. Median was chosen for this analysis, rather than mean, due to the smaller number of vocal/stimulation samples, and to exclude the effects of outliers (which can be seen in the error bars for some electrodes in figure 7 and 8). We have clarified this in the Methods on page 24.

(19) *“the most commonly observe stimulation effect” -> “observed”*

Thank you fixed.

(20) *“was performed by manually examination by a blinded reviewer” -> ?*

To determine the timing of the stimulation onset, we had a reviewer examine the frequency curves and select the time of any abrupt changes in the vocal frequency contour, if present. The reviewer was blinded as to the time stimulation was performed for any given vocal trial. We have clarified on page 23.

(21) “begasan” -> “began”

That is embarrassing , fixed. Thank you.

Reviewer #2

1.The authors do not reference figure 2c in the text.

Fixed, thank you.

2.Fig 3c: a fit would be helpful for this figure.

Thank you for the suggestion, a linear regression fit has been plotted.

3.Fig 4: How are the firing rates normalized here? This should be explained in the caption and I assume your Y axis Label should then change.

The firing rates of all PSTHs in the manuscript were spontaneous subtracted, but otherwise not normalized (consistent with our previous papers). Individual unit PSTHs were first calculated, then averaged to get the population PSTH. We have clarified this in the Methods (page 23). Additionally, in response to Reviewer 1, we have re-labeled all our PSTH y-axis as "Firing Rate Change".

4.For the 8.8% of units that displayed significant correlation coefficients: did you correct for multiple comparisons? If not, please do.

Yes, each unit's correlation coefficient was corrected for multiple comparisons, we have clarified in the Methods.

5. "These effects were stronger for units whose correlation was in the same direction as the expected vocal compensation" Please quantify.

Thank you for the suggestion. We now note in the Fig 4 legend (page 32), that for -2 ST and $r > 0$ (congruent correlation and behavior) 23 of 25 time bins over the first 250ms of feedback were significant vs. 3 for $r < 0$. For +2 ST it is 7 for $r > 0$ and 1 for $r < 0$.

6.Figure 7-8: I think the individual plots can be moved to a supplement.

We thank the reviewer for the suggestion, but we do not believe Nature Communications allows routine use of supplemental figures. Additionally, we feel that the concept that stimulating auditory cortex can rapidly change vocal production is a novel enough idea that a reader should be able to see the data themselves to evaluate the results.

7.Figure 7-8: it would be useful to summarize these plots into some kind of statistical analysis.

Figure 9A summarizes the stimulation effects for all electrodes, separately showing right and left hemispheres as well as their median stimulation effects, and those electrodes that were significant

(after correction for multiple comparisons). We have also now added an additional statistical analysis (mutli-variate regression) to look at the contributions of animal, hemisphere, row and column from Figure 7+8, which only showed a significant effect of hemisphere (page 9). Additionally, we have also now placed a marker for each plot in 7-8 to indicate statistical significance.

REVIEWERS' COMMENTS:

Reviewer #1 (Remarks to the Author):

Authors clarified all concerns and made improvements as suggested by reviewers. New descriptions look fair enough.

Reviewer #2 (Remarks to the Author):

The authors have addressed all my concerns. This is a great and unique addition to the literature!